# The Effect of Halliwick Method on Aquatic Skills of Children with Autism Spectrum Disorder

**DOI:** 10.3390/ijerph192316250

**Published:** 2022-12-05

**Authors:** Eliska Vodakova, Dimitrios Chatziioannou, Ondrej Jesina, Martin Kudlacek

**Affiliations:** 1Faculty of Physical Culture, Palacky University Olomouc, 771 11 Olomouc, Czech Republic; 2Faculty of Movement and Rehabilitation Sciences, KU Leuven, 3000 Leuven, Belgium

**Keywords:** adapted physical activity, therapeutic effect, gross motor skills, mental skills, aquatic training

## Abstract

Background: Swimming and the skills associated with participation in the aquatic environment tend to be an integral part of the movement literacy complex. Non-participation then affects the safety of movement in the aquatic environment and may also be the reason for the limitation of movement, psychological, and social development compared to peers. Methods: This study is a single-subject research study. The aim of this study is to evaluate the effect of a seven-week intervention program of the Halliwick method in the development of aquatic skills, gross motor skills, and mental skills relevant for aquatic competence for children with autism spectrum disorder. Seven children with autism spectrum disorder participated in swimming classes for a two-week baseline period and a seven-week intervention program of the Halliwick method, one time per week. To measure the effect in the field of aquatic skills, we used the Alyn Water Orientation Test 1. To determine the level of gross motor skills, we used the Gross Motor Function Measure test. Results: There was an improvement in aquatic skills and gross motor skills in seven participants; two of them did not improve in mental adjustment oriented to the breathing control sections in the water.

## 1. Introduction

Autism Spectrum Disorders (ASD) is a complex neurodevelopmental disorder characterized by difficulties in communication and social interactions. [1]. The Centers for Disease Control and Prevention (CDC) suggest that the diagnosis of ASD is increasing at a rate of 10–17% each year [2]. Several research studies were carried out in connection with intervention programs for children with ASD. The water’s buoyancy, turbulence, and resistance provide a feeling of freedom from the land’s constraints and allow the body’s joint free movement. Fundamental motor skills can also be improved when there is freedom of movement [3]. Elden [4] supports that water activities improve the sensory, cognitive, and physical abilities of children with ASD. Moreover, some studies tested swimming programs’ effects on swimming skills and gross motor function [5,6,7].

Children with ASD frequently have motor skills deficits that are present at a young age [8]. This negatively affects their participation in physical activity. In addition to delays in motor milestones, there are also deficits in gait, postural control, and motor planning [9]. Smith [10] emphasized the need for children with ASD to be encouraged to participate in games and other physical activities for motor skill development. Low levels of physical activity and high levels of sedentary behaviors play a significant role in their health consequences through childhood [11]. In addition, Sherrill [12] indicated that we can help individuals with ASD improve skills similar to these behaviors in different sports, such as swimming, by taking advantage of these stereotypical behaviors. For example, moving patterns such as moving their bodies backward and forward and their hands up and down can help them float and move independently in the water. Swimming and the skills associated with participation in the aquatic environment tend to be an integral part of physical literacy in its complexity [13]. Persons with ASD may be singled out from compulsory swimming lessons at school, but further participation in everyday life may be limited or assessed as risky. Non-participation then affects the safety of movement in the aquatic environment and may also be the reason for the limitation of movement, psychological, and social development compared to peers. Aquatic therapy or water-based activities are popular physical activities that encourage many individuals with ASD to participate in and enjoy the aquatic environment [14].

Exercising in the water reduces muscle spasms and pain, increases the range of motion of the joints, strengthens the body’s weak muscles, and improves blood circulation, lung function, balance, coordination, and posture [15]. According to the Autism Spectrum Disorder Foundation (ASDF), swimming and water activity can help children with ASD improve speech, coordination, social skills, self-esteem, and cognitive processing. While these children are often excluded from other sports (because of too many stimuli), being underwater can provide them with “quiet time alone”. Swimming provides an excellent opportunity for interaction at a level that is comfortable for children with ASD [16].

The Halliwick method can benefit everyone regardless of disability, but mainly focuses on individuals with physical or learning difficulties by aiding them in participating in water activities, moving independently in the water, and swimming [17]. The Halliwick method has been based on the principles of hydrostatics, hydrodynamics, and body mechanisms since its inception in 1949 [18]. It is a 10-point individual program that progressively teaches swimming skills from easy to more complex [19]. Usually, the Halliwick method is taught in group settings to motivate and favor social interaction and promote motivation between the group members. Its holistic characteristics and progressive content make it stand out from the other hydrotherapy methods in terms of the diversity of the people who can participate [20]. The literature indicates the beneficial effects of the Halliwick method in swimming skills on children with ASD. Some studies confirm the positive effect of programs using Halliwick’s methods in children with ASD [4,21,22,23], which was mainly about improving motor skills such as balance, dexterity, fine motor skills, flexibility, and, specifically, orientation in the water environment. The amount of stereotypical autistic movements (spinning, swinging, and delayed echolalia) decreases after ten weeks of the Halliwick program [21].

The aim of this study is to evaluate the effect of a seven-week intervention program of the Halliwick method in the development of aquatic skills, gross motor skills, and mental skills relevant for aquatic competences for children with autism spectrum disorder.

## 2. Materials and Methods

### 2.1. Study Design

This single-subject study concentrates on investigating the effect of the Halliwick method on aquatic skills in three different sections: mental adjustment, breathing control, and functional ability and gross motor function of children with autism spectrum disorder (ASD). The children’s aquatic skills were measured with the water orientation test Alyn 1 (WOTA-1), while their gross motor skills were measured with the gross motor function test (GMFM).

### 2.2. Intervention

Participants attended aquatic sessions with a duration of 60 min, 1 time per week for 9 weeks. The first two weeks served as the baseline period, and the remaining seven weeks served as the intervention period. The children performed the following swimming routine in a chlorine pool (28 °C) during the 2 week baseline period: 10-min warm-up including breathing exercises, diving exercises, jumping, and other such exercises, 30-min swimming training, which was kicking with kickboard, crawling arms, backstroke, and breaststroke, and a 10-min cool down, relax, and play session including diving activity and breathing activity.

The intervention period’s aquatic sessions, including a framework of the 10-point Halliwick method, taught the children basic swimming skills and independent movement in the water. These skills were introduced to them by specific exercises and game-based activities. The Halliwick method contains exercises for breathing control, balance to float in the water without assistance, transversal, sagittal, longitudinal, and combined rotation control, upthrust skills, and swimming stroke [24]. Since each child had a different skills level and different weaknesses, the program for each one was individually adapted.

### 2.3. Participants

The participants were selected based on their willingness to cooperate, as well as their legal representatives. These were persons arriving at the site of the planned research investigation. They were approached personally by the researchers. Based on inclusive criteria, all potential participants were approached. Out of the total number of nine participants, two participants were excluded from the study. One participant was excluded due to expressed disapproval of the legal representative. The other participant attended less than 75% of the sessions.

The group (Table 1) consisted of 6 males (88%) and 1 female (12%). Participants were aged 9.4 ± 2 years (mean ± standard deviation). The participants served on the single subject research design [25] for two weeks of baseline and seven weeks of intervention. All the children with ASD that participated in our study have already had previous experience in swimming classes. However, none of them had any experience with the Halliwick method prior to the study. The neuropsychiatrist and a clinical psychologist trained in the assessment of individuals according to the International Statistical Classification of Diseases and Related Health Problems (ICD-10) [26] performed the diagnosis of PAS participants.

All children had to meet the following inclusion criteria:diagnosed with ASD based on World Health Organization (childhood autism, atypical autism, Asperger syndrome) critieria,age between 7 and 12 years,ability to understand instructions adapted and implemented in accordance with the WOTA test battery,expressed consent of legal representatives,attended more than 75% of the sessions.

The children’s legal guardians were provided with a detailed explanation of the study, and all of them received and signed a written informed consent prior to the study’s launch. The study was approved by the Ethical Committee of the Faculty of Physical Culture Palacky University (n. r. 93/2021).

### 2.4. Measurements

The data for evaluating the swimming skills in the baseline and intervention period were collected in every session of the nine-week program. Gross motor skills were assessed during the first and the last aquatic sessions of the intervention period.

Aquatic skills were measured with the Water Orientation Test Alyn 1 (WOTA 1). The participants ended the aquatic session, and the instructors observed and filled in thirteen points of the WOTA 1 observation form. WOTA 1 was developed to evaluate how children with difficulties follow or understand instructions; WOTA 2 was developed to assess children with higher functional abilities [27]. It is mainly based on the mental adaptation, breathing control, and functional goal section of the Halliwick concept. Mental adaptation includes skills that evaluate the child’s adjustment to the properties of the aquatic environment: entering the pool willingly, side and back floating assisted by an instructor, and splashing water. The functional goal section includes skills that evaluate functional abilities of balance control: entering and exiting the swimming pool, side and back floating, maintaining a vertical position with straight arms, standing in water, holding the rope (a taut rope is strung across the pool–pool lane line), and sitting in the water and progressing along the pool edge using hands. Finally, the breathing control section included skills that evaluated the ability to control breathing: blowing bubbles in the water and submerging face in the water. There are 13 skill items on a 4-point ordinal scale [27]. The internal consistency and test–retest reliability of the WOTA 1 was satisfactory (α = 0.91; 0.94; 95% CI) [28]. The test WOTA showed itself to be reliable and have correlation to functional improvements on land [27]. Table 2 shows the sections of WOTA 1 test.

Gross motor skills were measured with the gross motor function measure test (GMFM). The participants individually carried out the test in the swimming pool area 1 h before their swimming session. The instructors explained each exercise verbally and with demonstration, and the maximum of three trials was permitted for each exercise. GMFM is an observational scale that assesses motor function and can quantify changes in gross motor ability in five dimensions; (i) lying and rolling, (ii) sitting, (iii) crawling and kneeling, (iv) standing, and (v) walking, running, and jumping [29]. The scores range from 0–3, where 0 = does not initiate task, 1 = initiates task (10%), 2 = partially completes task (10–99%), and 3 = completes task (100%). A higher GMFM score indicates better gross motor function. A percentage score is calculated for each dimension. Furthermore, the overall score can be calculated as the mean of the five-dimension scores [29]. In the present study, we used the overall scores of the five dimensions to evaluate the effect of the Halliwick method in the gross motor function at the start and the end of intervention period. Vascakova, Kudlacek, and Barrett [30] used the GMFM-66 test for 4 children with autism and found an improvement of 1.57% in their gross motor skills after 10 weeks of Halliwick swimming intervention.

To increase the reliability of the research, all the swimming sessions were conducted by the same two instructors. Both had previous experience in teaching adapted swimming and also had knowledge of the Halliwick method. A third person collected all the data during the aquatic sessions, later uploaded in a common excel file. The instructions for each activity were given to the children verbally and with demonstration. Furthermore, one instructor directly engaged with each child for more efficient teaching.

### 2.5. Analysis Procedure

Descriptive statistics were calculated for all outcome measures. Data from all participants who completed baseline, mid-, and post-intervention measurements and at least 75% of the swimming intervention were used. Missing data were recorded in the figures. In the figures, we record the number of sessions completed and the score achieved through the WOTA 1 test. There are 13 skill items on a 4-point ordinal scale. These 13 skills are divided into 3 structures: (a) general mental adjustment (max. 16 points), (b) breathing control (8 points), and (c) functional goals (36 points). Potentially, the highest possible score is the sum of (a), (b), and (c), which is 60.

We apply the GMFM at the beginning and at the end of the intervention. We present the test result as a percentage of the potential maximum in the test. The resulting score is the result of a comparison of both tests (pre, post).

Visual analysis using graphs and tables was used to determine if the intervention treatment was effective.

## 3. Results

Data revealed that the Halliwick method was effective in teaching aquatic skills to children with autism spectrum disorder. The most significant increase is between intervention 4 and 5, after which is the phase plateau.

Participant 1 (Figure 1) is male with atypical autism, 8 years old. He scored:

*Mental Adjustment:* 9 points from a maximum of 16. He entered the pool willingly with motivation from the instructor, but he had a problem with side floating, back floating, and immersing his ears in the water. We focused on mental adjustment skills. This participant refused to play games.

*Breathing Control:* 4 points from a maximum of 8. He started to blow bubbles through the mouth in the fifth session of intervention, he did not blow through the nose. He also did not submerge his head into the water during the program.

*Functional Abilities:* 29 points from a maximum of 36. He improved from the first session, entering the pool at the end of the sessions, but only with the supporting hands of the instructor, and exiting the pool without any support. He scored the maximum grade in the long arm hold, which was 10 s in a vertical position with support under hands, arms straight forward. He also scored the maximum grade in moving along the pool’s edge without support and standing and walking in the water without support. He also improved his balance skills, sitting in the water on the instructor’s thigh, chin in the water, with only a supported pelvis.

Participant 2 (Figure 2) is male with childhood autism, 7 years old. He scored:

*Mental Adjustment:* 11 points from a maximum of 16. He entered the pool willingly, with no need for motivation from the instructor. He had a problem with side floating and back floating, and he did not immerse his ears in the water. He improved in splashing water: he was able to splash the water around the face, but not on the face.

*Breathing Control:* 3 points from a maximum of 8. He started to blow bubbles through his mouth in the sixth session of intervention; he did not blow through the nose. He also did not submerge his head into the water during the sessions.

*Functional Abilities:* 31 points from a maximum of 36. His ability to enter the pool was the same from the beginning—he needed the instructor to support his forearms/upper arms—but he improved in exiting the pool, which he did independently. He increased in the long arm hold, which was a vertical position in the water with support under hands and standing, walking in the water, holding a rope, and sitting in the water.

Participant 3 (Figure 3) is male with Asperger syndrome, 11 years old. He scored the maximum grade in *Mental Adjustment*, *Breathing Control*, and *Functional Abilities* in the first session of intervention. He had a problem in baseline with side floating, so we worked on functional abilities, but he managed it. This participant needed more explanation of the activities. He participated in all games.

Participant 4 (Figure 4) is male with Asperger syndrome, 12 years old. He scored the maximum grade in *Mental Adjustment* and *Breathing Control* in the first session of baseline.

This participant had a problem with balance; he had a hard time with sitting in the water on the instructor’s thigh, and he also needed support in exiting the pool independently. We focused his program on this. Additionally, he scored the maximum grade in *Functional Abilities* in the sixth session of intervention.

Participant 5 (Figure 5) is male with Asperger syndrome, 9 years old. He scored:

*Mental Adjustment:* 15 points from a maximum of 16. He entered the pool willingly, and sometimes jumped. He had a problem with side floating; he needed head support but he immersed his ear into the water. He improved in back floating, and he did not object to floating with full support. Further, he splashed the water to others.

*Breathing Control:* 8 points from a maximum of 8. He blew bubbles through his mouth in baseline but he improved and started to breath with his nose in the fifth session of intervention. He had no problem with submerging his head and face in the water.

*Functional Abilities:* 35 points from a maximum of 36. He improved in entering the pool; at the beginning of the intervention he needed the instructor’s support hands only. He did not object to floating with full support back floating. He also improved in balance. He had a problem with sitting in the water: he needed support at the upper trunk sides at the beginning of the sessions. In the sixth session of the intervention, he needed only he mild support at the pelvis.

Participant 6 (Figure 6) is the only female in this study. She has childhood autism, 12 years old. She scored:

*Mental Adjustment:* 12 points from a maximum of 16. She slightly hesitant or was indifferent when getting into the water. She improved in back side floating in the fourth session of the intervention, until which she had refused to immerse her ear into the water—she was not relaxed and tried to get up. In the baseline, she was not able to perform splashing of the water, but in the fourth session of the intervention, she did not recoil from water around the face.

*Breathing Control:* 5 points from a maximum of 8. At the baseline, she did not put her mouth into the water, but she improved and, at the 3rd session of the intervention, she started to blow bubbles through her mouth but not through her nose.

*Functional Abilities:* 27 points from a maximum of 36. She needed support with the instructor’s forearms/upper arms in entering the pool. In the fifth session of the intervention she improved: she exited the pool without support and only with sitting. She improved also in intervention in the short and long arm hold and standing and walking in the water. Sitting in the water, she needed mild support around the waist. At the beginning, she needed full support at the upper waist and body.

Participant 7 (Figure 7) is male. He has atypical autism, 7 years old. He scored:

*Mental Adjustment:* 16 points from a maximum of 16. Since the baseline, he entered the pool willingly; he also did not recoil from water around the face from the baseline. He increased in side floating and back floating in the fourth session of intervention.

*Breathing Control:* 7 points from a maximum of 8. He did not blow bubbles through the nose, only through the mouth at all interventions. He also increased in submerging his head or face into the water.

*Functional Abilities:* 34 points from a maximum of 36. He increased significantly in the fourth session of the intervention. He immersed his ear in the water while side floating or back floating. He only needed support by holding the sides of the upper trunk; he did not need support of the head. He had a problem with the short and long arm hold, but in the fourth session of the intervention, he improved and was able to do it only with hand support. He also improved his balance skills in sitting in the water by the fourth intervention.

Figure 8 shows the comparison of all participants in the total score of the WOTA 1 test. The highest possible score is 60, and the value of the individual scores of the participants is the median of the total score in individual measurements over nine weeks. Score 47 marks the median of all results together. We observe that participants 1, 2, and 6 did not achieve the median score. Participants 3, 4, and 5 scored higher than the calculated median. These were the participants with Asperger syndrome. In Table 3 there are nine-week session outcomes via the WOTA 1.

In our study, one of the purposes was to investigate the effect of an aquatic intervention on the gross motor function and aquatic skills of children with ASD. GMFM value, which represents the motor function, after seven weeks of aquatic intervention for children with ASD presented improvement. These results of total GMFM score are presented in (Table 4). Participant 2 had the most significant improvement. He mainly improved his balance skills.

We found no significant correlation between GMFM and WOTA scores. Nevertheless, it is possible to trace improvements in both test techniques in all participants. Participant 7 had a significant improvement in the WOTA test (14 points; starting 43 and final 57) and the smallest (3.5%) improvement in GMFM among all participants.

Taking a closer look at the individual domains of the GMFM test, we achieved different variability of improvement in all participants. The participants achieved the greatest improvement in the domain “D: Standing”, in the specific item “balance” (item 57, 58-STD: Lifts L and R, arms free, 10 s).

## 4. Discussion

This study investigated the effect of the Halliwick method on aquatic skills in children with ASD. Our study showed that a seven-week specific program of the Halliwick method could improve functional ability, mental adaptation, and breathing control in water and could also improve gross motor function. The results of the study were analyzed using graphical representation. The Halliwick method is particularly suitable for individuals with disabilities and individuals who do not have much experience with water [17,31]. Our study confirms that children who already have high functional ability with mental adjustment cannot benefit from the Halliwick method as much as those with little or no experience.

As noted in the results, there were different changes in the outcomes among participants in this study. Children with childhood and atypical autism showed lower scores at baseline, but two of them had the biggest improvement. Participant 6, with childhood autism, had the biggest improvement, from 19 to 44 in the WOTA scale; she was the only female in this study. The second best improvement was observed in participant 1 with atypical autism, whose score reached from 20 to 38. On the contrary, Participants 3 and 4, with Asperger syndrome, already scored high on the initial WOTA at the beginning, so we observed the ceiling effect in the measurement of our intervention. They demonstrated less change, and their maximum score remained for the rest of the program. One of these children did not reach the minimum detectable change, because his score was high at the beginning of the program. Pan [32] investigated the effectiveness of water exercise on aquatic skills of boys with Asperger syndrome. They indicated significant improvement and a decrease in social behaviors.

We see the best improvement in most participants between intervention 4 and 5. After that, there are only minimal changes. This may be because the program has the greatest benefit at the beginning of the program, or some individuals reach their ceiling effect. Even so, participants can still benefit from the aquatic environment. However, there is an opportunity to focus on other procedures, repetition, or deepening of learned movement elements. Further in this phase of improvement, there is the possibility of involving the family and visiting the swimming pool. Direct therapeutic service is not necessary here [14,22,24,33].

The highest improvement in participants 1 and 6 was observed at the start of the therapeutic program. The results may be due to the fact that swimming programs begin with mental adjustment, and only then are more challenging tasks included, such as rotation and balance. Some individuals with ASD need more time to adapt—they have difficulty changing activities. Our testing always took place at the end of the lesson. Aquatic therapy is based on experience with water in children with disabilities. Gradual adaptation to the water environment was very important for these participants. When they arrived at the third session, they already knew what kind of environment they were going to; the instructor was also better prepared for them and was able to adjust the conditions and create individual modifications for the given participant. For example, the light might need to be dimmed or the noise in the surroundings reduced.

It is important to mention that the most challenging skills were the side and back floating with only the sides of upper trunk being held by instructor. These skills are part of the mental adjustment and functional ability section and caused the most fluctuations in results. Furthermore, two participants with ASD who could not blow bubbles with their head immersed in the water or retrieve an object from the pool´s bottom never succesfully put their head in the water. That leads to the conclusion that even though we saw significant improvement in the children´s swimming skills, there are some barriers we could not overcome. Perhaps we needed more time with these participants. The present study results are supported by previous studies that also pointed out improvement in the aquatic skills and motor development [4,21,22,23,24,30] of children with ASD through the Halliwick method.

The issues of ASD and the use of aquatic therapy were addressed by the authors Huetting, Yilmaz, and Pan, who, unlike in our study, used the HAAR test battery, also based on Halliwick’s concept. This battery contains very similar test items to the WOTA 1 we used. They showed improvement in the areas of water orientation skills, breathing skills, floating skills, balance, speed, agility, and power scores after 10 weeks of program. All the listed authors also focused on the age category of up to 12 years [14,21,34]. In addition to the effect of aquatic therapy on motor skills and aquatic skills, the Pan and Yilmaz study focused on reducing behavioral problems, and the Yilmaz study showed that stereotyped movements (spinning, rocking, and delayed echolalia) were reduced in autistic people after aquatic therapy. Another possible cause of the reduction in antisocial behavior may be due to a favorable response to individualized instruction and positive change. The instructor must be patient.

From our point of view, we observed that the stereotyped movements sometimes made it difficult to teach some skills. For example, participant 2 always wanted to hold an object in his hands. We put an object on the other side of the pool and demonstrated different exercises. In addition, we noticed that Yilmaz [23] used the constant time delay technique to teach the Halliwick swimming rotation skills for children with ASD. This could potentially be used in other research with children who do not want to put their head into the water.

All participants improved in gross motor skills. The best improvement was achieved by participant 2, with childhood autism; his improvement in the total GMFM score was 12.7%. On the contrary, participant 7 achieved the smallest improvement, by 3.5%, but this participant achieved a jump in functional abilities in the aquatic environment after his illness, between three and four sessions. This may be due to the repetition of the given exercises, which the child could have fixed during the break between interventions as part of motor learning, but a number of other factors play a role here, for example, a change in medication. Previous studies have also used a combination of WOTA and GMFM tests to evaluate the effect of water therapy. Their use was addressed exclusively in participants with cerebral palsy [35,36]

Of great importance is perceived quality of life and social behaviors of children with ASD. In accordance with some [37], we generally consider the water environment and activities in water to be highly suitable for increasing the subjective perception of the quality of life and for the development of their objectively measurable structures. Our findings confirm the effect of a several-week program with a demonstrable effect on the development of competencies, which we can consider, in accordance with other findings, as an assumption of an increase in the quality of life. Few studies have also investigated the effect of hydrotherapy on the social interactions and behaviors of children with ASDs, not only motor development [38]. The search for connections between these factors and the effect of specific swimming programs can be part of other qualitative research.

### Strenght and Limits of the Study

There are several limitations to this study. The first of these was finishing the final phase of the single subject design because of the COVID-19 situation in Czech Republic. Second, one of the instructors could not speak the Czech language, and communication was challenging. Ideally, future studies would include instructors and participants who speak and understand the same language. The third limitation was the relatively small sample size. This point encourages future studies with a larger sample size. The findings cannot be generalized due to its small sample size and single subject design.

## 5. Conclusions

The purpose of the current study was to evaluate the effect of a seven-week program of the Halliwick method on aquatic skills for children with autism spectrum disorder. The study provides initial evidence in favor of the Halliwick method and suggests it as a complementary method to improve aquatic skills in all three sections: mental adjustment, breathing control, and functional ability of the WOTA 1 test, and also in motor function in ASD. The Halliwick method effectively increased swimming skills and functional abilities in children, especially those with low experience with water. Children, already diagnosed with Asperger syndrome, who scored better results in the first measurement of the WOTA test did not achieve such a degree of improvement as children with worse initial data. Seven participants improved their aquatic skills with only seven weeks of intervention. Each child had only one hour of aquatic session per week. We also noted improvement in all children in the GMFM tests, with the lowest improvement of 3.5% (atypical autism) and the highest of 12.7% (childhood autism).

Except for the type of Asperger’s syndrome, we noticed a leap improvement in most of the participants right after the first training unit with the content of Halliwick’s methodology. The highest rate of improvement was achieved between week 4 and 5 of the intervention, most often followed by a plateau phase. Our findings can form a basis for future large-scale studies.

## Figures and Tables

**Figure 1 ijerph-19-16250-f001:**
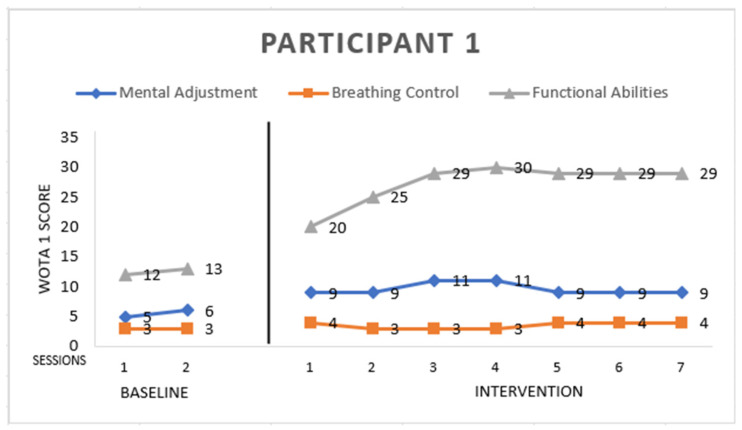
Highest possible scores: Mental Adjustment = 16, Breathing Control = 8, Functional Abilities = 36.

**Figure 2 ijerph-19-16250-f002:**
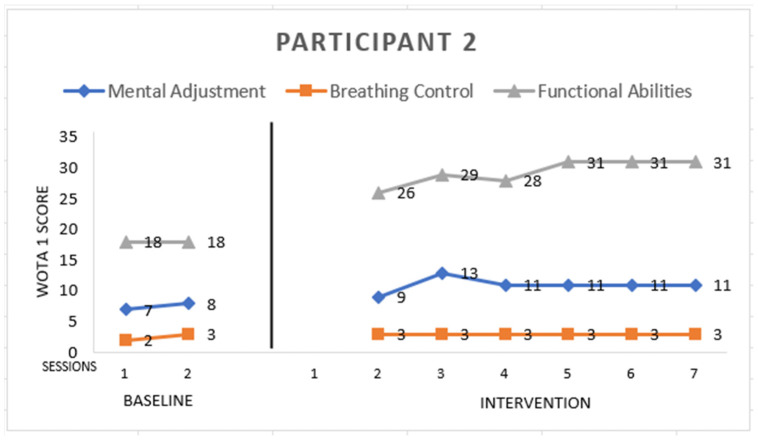
Highest possible scores: Mental Adjustment = 16, Breathing Control = 8, Functional Abilities = 36.

**Figure 3 ijerph-19-16250-f003:**
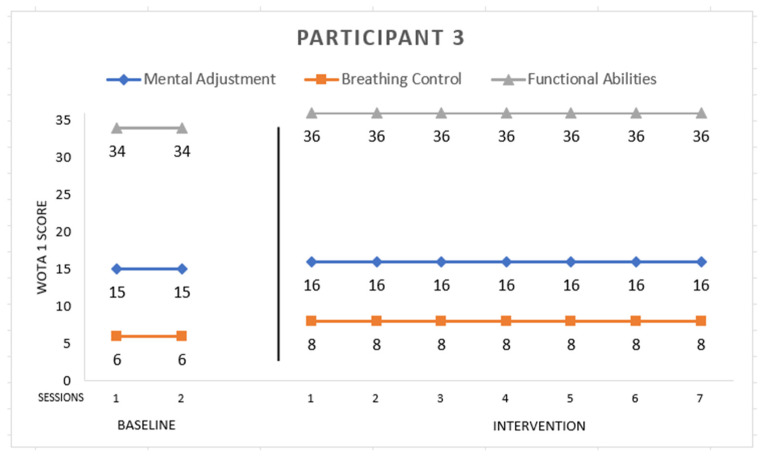
Highest possible scores: Mental Adjustment = 16, Breathing Control = 8, Functional Abilities = 36.

**Figure 4 ijerph-19-16250-f004:**
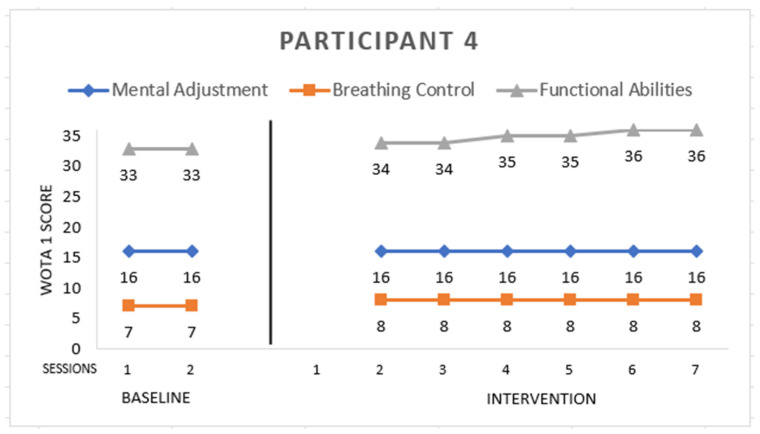
Highest possible scores: Mental Adjustment = 16, Breathing Control = 8, Functional Abilities = 36.

**Figure 5 ijerph-19-16250-f005:**
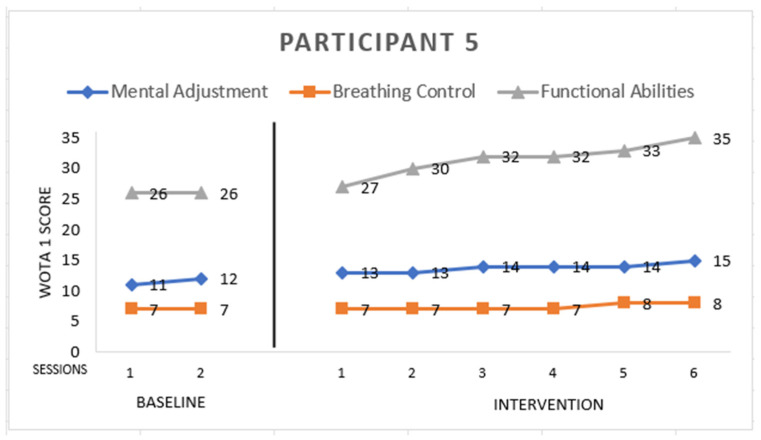
Highest possible scores: Mental Adjustment = 16, Breathing Control = 8, Functional Abilities = 36.

**Figure 6 ijerph-19-16250-f006:**
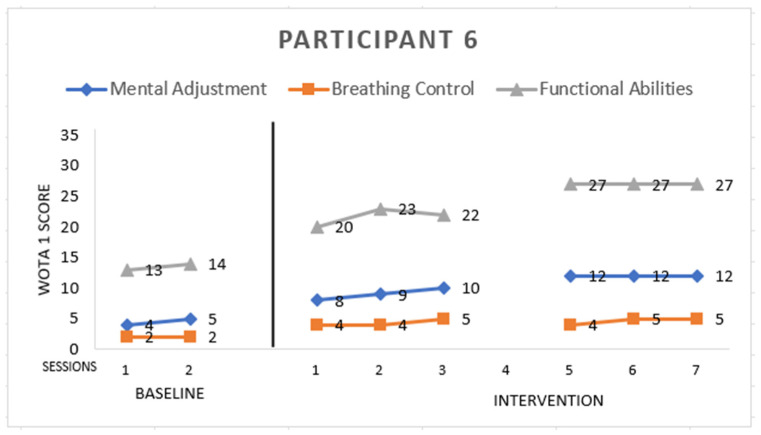
Highest possible scores: Mental Adjustment = 16, Breathing Control = 8, Functional Abilities = 36.

**Figure 7 ijerph-19-16250-f007:**
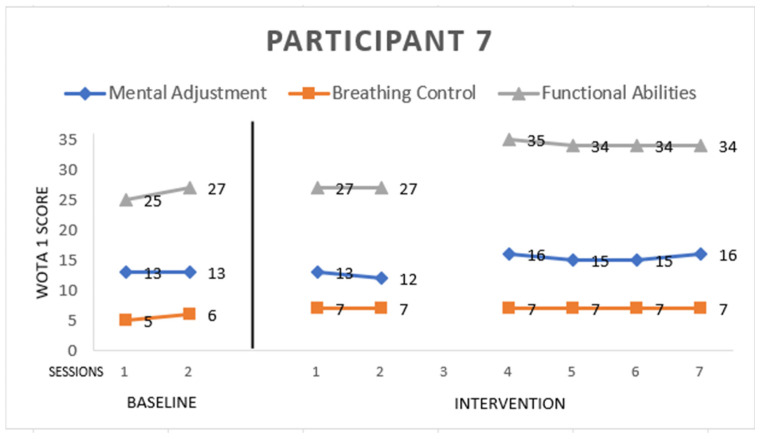
Highest possible scores: Mental Adjustment = 16, Breathing Control = 8, Functional Abilities = 36.

**Figure 8 ijerph-19-16250-f008:**
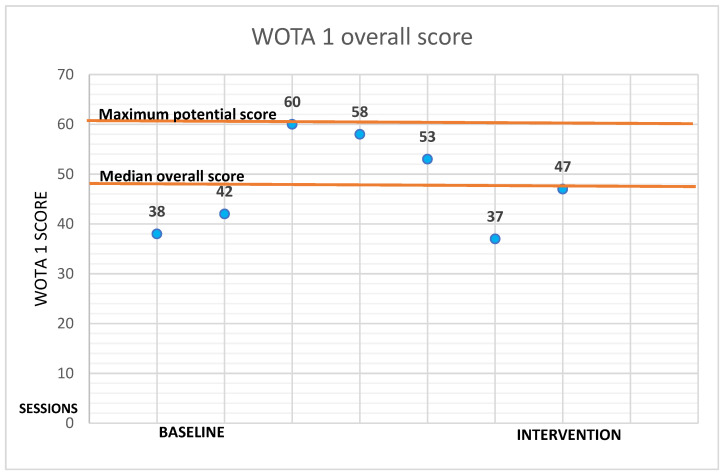
Maximum potential score = 60; median scores of all overall participant scores = 47. P1 = Participant 1, P2 = Participant 2, P3 = Participant 3, P4 = Participant 4, P5 = Participant 5, P6 = Participant 16, P7 = Participant 7.

**Table 1 ijerph-19-16250-t001:** Research subject by gender, age, and type of autism.

P	M/F	Age	Type of Autism	Code of Diagnosis
**1**	M	8	Atypical autism	F 84.1
**2**	M	7	Childhood autism	F 84.0
**3**	M	11	Asperger syndrome	F 84.5
**4**	M	12	Asperger syndrome	F 84.5
**5**	M	9	Asperger syndrome	F 84.5
**6**	F	12	Childhood autism	F 84.0
**7**	M	7	Atypical autism	F 84.1

P = Participant; M = Male; F = Female; The type of autism was evaluated based on classification in World Health Organization (ICD-10).

**Table 2 ijerph-19-16250-t002:** WOTA 1 sections of testing [27].

**A. General Mental Adjustment**	General mental adjustment
	Splashing water
	Side and back floating with instructor’s help
**B. Breathing Control**	Bubbles
	Submerging
**C. Functional Goal**	Entering and exiting pool from pool edge
	Side and back floating with instructor’s help
	Short and long arm hold
	Progression along pool edge using hands
	Standing in the water
	Holding rope
	Sitting in the water

**Table 3 ijerph-19-16250-t003:** Nine-week session outcomes via the WOTA 1 [27].

Participant	WOTA-1ST SCORE	WOTA-9TH SCORE	WOTA Changes in Scores	Greater Than MDC?
1	20	38	18	Yes
2	27	45	18	Yes
3	55	60	5	Yes
4	56	60	4	No
5	44	58	14	Yes
6	19	44	25	Yes
7	43	57	14	Yes

MDC for WOTA 1 = 4.2. MDC = minimal detectable change, WOTA = water orientation test.

**Table 4 ijerph-19-16250-t004:** GMFM total scores for before and after intervention (%).

P	Before (%)	After (%)	Deference (%)
**1**	72.6	81.9	9.3
**2**	57.3	70	12.7
**3**	89.9	98	8.1
**4**	92.3	98	5.7
**5**	93.3	99	5.7
**6**	62.1	68	5.9
**7**	63	66.5	3.5

P = Participant. Before (%) = Score by percentage before intervention; After (%) = Score by percentage after intervention; Deference (%) = Difference in scores by percentage.

## Data Availability

Data are with the lead author and first co-author. They are not available in any public repository.

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
