# Peer review of "The Effect of Halliwick Method on Aquatic Skills of Children with Autism Spectrum Disorder"

_ijerph, 2022, doi:10.3390/ijerph192316250_

Round 1

Reviewer 1 Report

This manuscript analyzed the aquatic skills and gross motor function skill in children with ASD after aquatic exercise intervention using Halliwick Method. The topic is interesting and a subject that needs more research. However, there are several problems needing attention.

Introduction

Page 2, 3rd paragraph, the authors state “From its inception in 1949…” There is a sentence structure error. Please restate.

Page 2, 3rd paragraph, the authors stated “The results showed a significant increase in correct target…”. Please elaborate more on “correct target skills” and provide more literature and evidence for Halliwick method.

Page 2, 4th paragraph, please add a comma after “gross motor skills”

Participants

The participants information should go under “Materials and Methods” 

The authors stated “One participant due to denial of the parent to participate in the study…” The sentence is incomplete. Please restate. 

“Participants were aged 9,4+2” should be “9.4+2” (not a comma)

The authors stated “The diagnosis of ASD by neuropsychiatrist and a clinical….” There is a sentence structure error. Please restate.

Inclusion criteria

Please elaborate on “correctly registered”.

Materials and Methods

Please add more detailed sampling methods under “Participants” section. How did you recruit your participants?

It may be appropriate to have “Intervention” subheading section followed by “The children performed the following swimming routine in a chlorine pool”

Authors provided detailed information on each period. Under “Intervention” section, please provide the overall intervention protocol first then explain each period.

Measurements

The authors stated “…standing in water, holding the rope, …”. What is the rope? Are you talking about a pool lane line?

The authors stated “Each item is scored using a 4-point ordinal scale, it has the total of 13 skills that are easy to measure and the maximum score to be obtained is fifty-two.” I suggest the following statement  “There are 13 skill items on a 4-point ordinal scale with the highest possible score of 52.”

The authors stated “It is showed that WOTA 1 has very good values of internal consistency (Cronbach α= 0.91) and test-retest reliability (ICC= 0.94; 95%CI).” I suggest the following statement “The internal consistency and test-retest reliability of the WOTA 1 was satisfactory (α= 0.91; 0.94, 95%CI)

Please be consistent with acronym use for gross motor function measure test with GMFM. For example “Gross motor skills were measured with the gross motor function measure test.” The authors spelled out this sentence but later use “GMFM”

Please add the analysis procedure.

Results

Please provide the overall results of 7 participants and the participation rate.

The author stated, “The most significant increase is between interventions 4-5, then is phase plateau.” I suggest the following statement “The most increase was observed among all participants during interventions 4-5.”

For each participant’s reporting, the statement “He reached the scores in” is strange. Please restate.

The authors provided in-depths results of each measurement. I would like to see how each participant performed and progressed Halliwick method/swim sessions in order to make this current manuscript stronger.

Figures

All figures are very clear.

Discussion

The authors stated “Our study shows that children who have already a lot of experience with swimming and high functional ability with mental adjustment cannot benefit from Halliwick method as those with little to no experience.” However, the current manuscript did not compare experienced swimmer and no swim experience. Additional analysis is needed for this statement. If you are not comparing based on their experience, please rephrase this statement.

The authors stated, “Furthermore we observed that all children could already enter the pool independently or with minimal support, and so they were better with tasks related to functional abilities.” I do not see the relevance of entering the pool independently with “better with tasks related to functional abilities.” Please add more explanation with previous literature on this topic.

Overall the discussion section lacks with flow of the explanation. I encourage the authors to continue working on the flow of the Discussion to strengthen and make the manuscript clear relevant to your findings and previous literature.

Conclusion

I suggest omitting the statement “it is unclear if the findings can be positively generalized for a specific type of an individual.” It is clear that this finding cannot be generalized due to its small sample size, and single case design.

Reference

Please make sure to check the reference format to be consistent with the format. (Int. J. Environ. Res. Public Health).

·         Reference 12, and 17 are in APA format.

·         Some journals are abbreviated, and some are spelled out.

Author Response

1) The Halliwick method is based on the principles of hydrostatics, hydrodynamics, and body mechanisms, since its inception in 1949.

  • accept

2) correct target skills - 

Some studies confirm the positive effect of programs using Halliwick's methods in children with ASD. [4,21-23]. It was mainly about improving motor skills such as balance, dexterity, fine motor skills, flexibility and specifically orientation in the water environment. The amount of stereotypical sutistic movements (spinning, swingind and delayed echolalia) decrease after ten weeks of Halliwick program [21].

  • accepted and rewritten 

3) Page 2,4th paragraph (comma)

  • accepted

4) "Participants"

  • accepted all

5) Inclusion criteria - rewritten - expressed consent of legal representatives; The children’s legal guardians were given a detailed explanation of the study and, all of them received and signed a written informed consent prior to the study's launch. The study was approved by the Ethical Committee of the Faculty of Physical Culture Palacky University (n. r. 93/2021).

  • accepted

6) Sampling methods - It was a deliberate selection of participants. The participants were selected based on their willingness to cooperate, as well as their legal representatives. These were persons arriving at the site of the planned research investigation. They were approached personally by the researchers. Based on inclusive criteria, all potential participants were approached.

  • accepted and add

7) Intervention (subheading) 

  • accepted

8) Intervention - we restructured parts according the reviewer. Information (moved to "Measurements") - The data for evaluating the swimming skills in the baseline and intervention period were collected in every session of the nine-week program. Gross motor skills were assessed during the first and the last aquatic sessions of the intervention period.

  • accepted

9) Rope - added - a taut rope is strung across the pool - pool lane line.

  • accepted

10)  Ordinal scale - rewritten.

  • accepted

11) WOTA validation - rewritten.

  • accepted

12) Acronym GMGM 

  • accepted

13) Analysis procedure

  • accepted

14) Result - overall results

  • accepted and added

15) "reached the scores in" - we change to "scored"

  • accepted

16) "I would like to see how easch participant perforemd and progressed H.m...."

  • accepted and added in table 3

17) Discussion

  • all accepted and we tried to included all reminders

18) Conclusion

  • accepted

19) Reference

  • accepted

Reviewer 2 Report

I found one major issue, which is related to the use of the WOTA 1 assessment. WOTA 1 has 13 items, each of them can be rated with 4 points maximum. This gives a maximal total score of 52, as has been explained in the text. However, in this manuscript the maximum WOTA score is: 16 for mental adjustment + 8 for breath control + 36 for functional abilities = 60

With this, WOTA outcomes are unreliable: authors miscalculated or added 2 items?

Apart from this major issue I have questions that would need to be addressed.

1. Participants

-       Male and female instead of men and women

-       Why the WHO classification has been used and not the DSM-5?

-       Ability to understand instructions: verbal instructions were given according to the measurements sections.

o   Level of communication and social interaction is important in ASD, why hasn’t an instrument like ADOS been used?

Children were able to follow verbal instructions. WOTA 1 is a simple form of WOTA 2 and has been developed for children with difficulties in understanding and following instructions. Why WOTA 2 has not been used?

2. Study design

The study focused on 3 sections: mental adjustment, breathing control, and functional ability. An table of which of the 13 items belongs to the 3 sections might help readers to understand WOTA1.

The children were divided into four different swimming groups: seven children in four groups.

-       How as the distribution? Were these groups with also other children? How did the customized intervention take place?

The two baseline sessions contained: 10-min warm-up including breathing exercises, diving exercises, jumping and other such exercises, 30-min swimming training was kicking with kickboard, crawling arms, backstroke and breaststroke and 10-min cool down, relax and play including diving activity and breathing activity.

-       How does this match with the poor WOTA scores after session 1 and 2 of e.g. #1 and #6? This would have been a point for the discussion section.

GMFM

This assessment has 5 domains, in which ones the changes were observed. 

-       How could they have correlated with changes in WOTA 1? Were the changes clinically important enough to exceed the minimum clinical important difference?

Results

The graphs of the 7 participants show “jumps” in the functional ability section:

-       #1 changes from 13 to 20 between session 2 and 3

-       #2 changes from 18 to 26 between session 2 and 3

-       #6 changes from 14 to 20 between session 2 and 3

-       #5 changes from 22 to 27 after not having attended in week 4

-       #7 changes from 27 – 35 after having not attended in week 3

o   Why? This should have been discussed in the discussion section

Discussion:

Authors write that: It is important to mention that the most challenging skills was side floating and back floating without support of instructor. However: WOTA 1 states at #5 “side floating with instructor’s help” that the maximal of 4 points are gained when: “Support the sides of: pelvis/waist/upper trunk - initiates floating (ear is immersed) and returns to vertical position”. 

When authors rate unsupported side float with 4 points, the rating isn’t reliable as far as it concerns this point and have to be recalculated.

·      The same counts for back floating

WOTA results are clearly different for participants. The 3 participants diagnosed with Asperger syndrome had high scores, while in 3 of the other participants scores were quite low, esp #1 and #6. This might have been explained.

In general: language should be checked!

Author Response

1) WOTA score (major issue)

  • accepted and corrected

2) Male and female

  • accepted

3) WHO vs. DSM 5: In the European context, the WHO categorization is used more often. Considering the fact that we still do not have an official translation of the 11th revision in the Czech Republic, the 10th revision is used in our terms and conditions.

4) Ability to understand instructions...

  • Accepted and changed because we incorrectly formulated the original inclusive criteria.
  • We didn´t use the ADOS, we were not forced to use this procedure in testing. Despite the fact that we did not have a trained specialist with relevant education (psychologist, psychiatrist).
  • WOTA: We gave priority to the WOTA version over the WOTA 2 version mainly because it is a simpler version that can be used especially by children with communication difficulties. The second important reason was the fact that the testing took place after each session, and therefore, for organizational and functional reasons, a 13-item test battery appeared considerably more suitable than a 27-item one.

5) Study design - 3 sections

  • accepted see table 2

6) Study design and dividing of the children - accepted, we delated (it was irelevant, we did not divide them into groups. Individual teaching took place and the division was only time-based, taking into account that the session for some children took place at the same time, but with two examiners (at the same time the authors of this article).

7) GMFM - we didn´t found the significant correlation between WOTA and GMFM. Nevertheless, we expanded the text and specified the results according to the reviewer's comments.

  • added 

8) Results / discussion (note about "jump")

  • we've added a larger section of text to the discussion section that explains

We see the best improvement in most participants between 4-5 interventions. After that, there are only minimal changes. This may be because the program has the greatest benefit at the beginning of the program, or some individuals reach their ceiling effect. even so, participants can still benefit from the aquatic environment. But there is an opportunity to focus on other procedures, repetition or deepening of learned movement elements, further in this phase of improvement there is the possibility of involving the family, visiting the swimming pool, direct therapeutic service is not necessary here. [14,26,34 and Prupas]

We observe in participants 1 and 6, the highest jumps in improvement at the start of the therapeutic program. The results may be due to the fact that swimming programs begin with mental adjustment, only then are more challenging tasks included, such as rotation, balance. Some individuals with ASD need more time to adapt, they have difficulty changing activities. Our testing always took place at the end of the lesson. Aquatic therapy is based on experience with water in children with disabilities. Gradual adaptation to the water environment was very important for these participants, when they arrived at the 3rd session they already knew what kind of environment they were going to, the instructor was also better prepared for them and was able to adjust the conditions and create individual modifications so that the given participant. For example, the light had to be dimmed, the noise in the surroundings reduced.

9) Discussion - Side floating and Back floating - Thank you for the reminder, as we incorrectly stated the level of instructor support during the translation.

  • accepted and changed

10) Discussion - we correct all the discussion according to your reminders.

  • accepted and corrected

Round 2

Reviewer 1 Report

It was a pleasure to review this manuscript. The manuscript is markedly improved in terms of rigor and clarity. 

Reviewer 2 Report

thanks for the explanations